# Carcinoid Crisis: A Misunderstood and Unrecognized Oncological Emergency

**DOI:** 10.3390/cancers14030662

**Published:** 2022-01-28

**Authors:** Camilla Bardasi, Stefania Benatti, Gabriele Luppi, Ingrid Garajovà, Federico Piacentini, Massimo Dominici, Fabio Gelsomino

**Affiliations:** 1Department of Oncology and Hematology, Division of Oncology, University Hospital of Modena, 41124 Modena, Italy; camilla.bardasi@gmail.com (C.B.); stefania.benatti@unimore.it (S.B.); gabriele.luppi1@gmail.com (G.L.); federico.piacentini@unimore.it (F.P.); mdominici@unimore.it (M.D.); 2Medical Oncology Unit, University Hospital of Parma, 43100 Parma, Italy; ingegarajova@gmail.com

**Keywords:** crcinoid crisis, neuroendocrine tumors, hemodynamic instability, octreotide

## Abstract

**Simple Summary:**

In this review, the Authors are going to discuss the main highlights of the Carcinoid Crisis, an uncommon manifestation related to neuroendocrine tumors, focusing on the potential etiopathogenetic mechanisms, clinical implications, potential treatments and prophylaxis.

**Abstract:**

Carcinoid Crisis represents a rare and extremely dangerous manifestation that can occur in patients with Neuroendocrine Tumors (NETs). It is characterized by a sudden onset of hemodynamic instability, sometimes associated with the classical symptoms of carcinoid syndrome, such as bronchospasm and flushing. Carcinoid Crisis seems to be caused by a massive release of vasoactive substances, typically produced by neuroendocrine cells, and can emerge after abdominal procedures, but also spontaneously in rare instances. To date, there are no empirically derived guidelines for the management of this cancer-related medical emergency, and the available evidence essentially comes from single-case reports or dated small retrospective series. A transfer to the Intensive Care Unit may be necessary during the acute setting, when the severe hypotension becomes unresponsive to standard practices, such as volemic filling and the infusion of vasopressor therapy. The only effective strategy is represented by prevention. The administration of octreotide, anxiolytic and antihistaminic agents represents the current treatment approach to avoid hormone release and prevent major complications. However, no standard protocols are available, resulting in great variability in terms of schedules, doses, ways of administration and timing of prophylactic treatments.

## 1. Introduction

Neuroendocrine Tumors (NETs) are a heterogeneous family of neoplasms that can arise from any district of the body and can occur with an extremely wide range of symptoms and clinical manifestations, due to hormonal secretion. Carcinoid Syndrome (CS) is the most typical and common clinical presentation in functional NETs and is characterized by diarrhea, gastrointestinal discomforts, such as cramps and nausea, facial flushing with apparent peripheral cyanosis, eventual right-sided valvular heart disease preceded by palpitation and dyspnea with bronchospasm. Very rarely, patients with NETs can also exhibit a life-threatening occurrence known as Carcinoid Crisis (CC), generally described as a sudden onset of hemodynamic instability (prolonged hypertension or severe hypotension, unresponsive to standard practices), sometimes accompanied by characteristics of carcinoid syndrome, such as prolonged flushing, wheezing and hyperthermia. The underlying mechanism of CC is still not well known, but some Authors first hypothesized that CC could represent an extreme complication of the Carcinoid Syndrome caused by a sudden and massive release of tumor hormones that may be triggered by tumor manipulation or anesthesia [1,2].

CC is mostly associated with foregut (respiratory tract, thymus, stomach, duodenum and pancreas) and midgut (small intestine, appendix and right colon) NETs [3]. Although theoretically every kind of tumor stress can cause CC, it typically occurs during invasive procedures, such as surgery and liver embolization [4], but it can also arise during clinical examination, biopsy, mammography [5], transesophageal echocardiography [6] or induction of anesthesia. Some cases of spontaneous onset have also been described [2,7,8]. The reported incidence of CC is about 7% in patients with NETs [9] undergoing abdominal surgery [10,11], but more recent works observed a higher number of cases, with a maximum incidence of 24–30% [12].

## 2. Aim

In this review, the Authors are going to discuss this uncommon manifestation, focusing on the potential etiopathogenetic mechanisms, clinical implications and potential treatments and prophylaxis.

## 3. Materials and Methods

The database PubMed was searched for the characteristics of CC using the term “carcinoid” combined with “crises” and “crisis” for publications with English abstracts up to September 2021. “Malignant carcinoid syndrome” was not considered. Studies included: (1) original articles, case series or case reports; (2) reporting at least one of the characteristics of Carcinoid Crisis (hemodynamic instability, continuous flushing, tachycardia predisposing to arrhythmias, bronchial wheezing, hyperthermia, peripheral cyanosis, severe diarrhea, central nervous system dysfunction with coma). In the section “Clinical Definition and Presentation”, the Authors identified 12 case reports concerning CC.

## 4. Clinical Definition and Presentation

Carcinoid Crisis is an extremely rare life-threatening event, with little data published on this topic. The first report of CC dates back to 1964 [2], when Kahil et al. described a case of a 41-year-old woman who underwent surgical resection of a NET of the ileum and a few months later started to manifest episodes of flushing, cramps and pruritus, interpreted as malignant CS. She was treated with a peripheral serotonin antagonist, cyproheptadine and was trained to adopt a low-tryptophan diet, obtaining good control of symptoms. Thirteen days after the discontinuation of therapies, the woman manifested a sudden onset of apprehension, oppressive chest pain, abdominal cramps, frequent diarrhea, facial flushing, pruritus, paresthesia and hyperaesthesia, peripheral vascular collapse with pale cold cyanotic extremities and progressive extreme hypotension. Neither metaraminol, a potent vasoconstrictor, nor norepinephrine, was effective in blood pressure control. One hour after the onset of symptoms, a single intravenous injection of cyproheptadine was administered with a dramatic cessation of the thoracic and abdominal pain. The Authors named this an extremely dangerous condition of Carcinoid Crisis, thus suggesting for the first time that it could represent a serious complication of CS, caused by a sudden release of active substances by neuroendocrine cells, provoked by stress on tumor masses.

Table 1 summarizes published case reports of CC.

As assumed by Kahil et al., in the absence of external stress, neuroendocrine tumor cells produce a share of vasoactive peptides that provoke CS. Instead, when a greater stimulus takes place, a hormonal storm can trigger CC. As shown in Table 1, the most commonly reported primary tumor locations that can cause CC are lung and small bowel (ileum), which also represent the most common sites associated with CS, because of the overall major release of vasoactive peptides compared to other districts [3].

As previously shown in Table 1, not only direct manipulations of tumor mass, such as bronchoscopy [16,20], liver biopsy [13,16] or locoregional treatments [18,21], but also other kinds of tumor solicitations, such as the induction of anesthesia [14] or the infusion of radiotracers [17], can contribute to the rapid onset of CC. In addition, following the results of the NETTER-1 trial, the increasing use of peptide radionuclide therapy (PRRT) has led to a rise in CC due to tumor lysis [22], most frequently after the first cycle of treatment, and the main risk factors include large tumor burden, liver metastases, previous CS, carcinoid heart disease, advanced patient age, high chromogranin A levels and high 5-HIAA levels. Furthermore, in some cases, PRRT-induced CC can also occur long after therapy [24].

Furthermore, clinical presentation is remarkably variable. For instance, Kromas et al., described a case of a 31-year-old woman who presented chest pain, newly onset asthma and a sudden onset of hypotension and wheezing during rigid bronchoscopy [20]. Koopmans et al. reported the case of a 61-year-old woman who developed vomiting, accompanied by flushing, edema and severe hypotension during an ^18F^-DOPA PET scan [17]. The explanation proposed by the Canadian group of Seymour et al. is that each tumor secretes a specific cocktail of hormones, which leads to discrepancy regarding recognition and symptoms and contributes to the great clinical variety of CC [25].To date, there has been no international consensus on the most appropriate definition of CC. Clinicians continue to identify CC as a rapid onset of hemodynamic instability, unresponsive to conventional management associated with characteristics of CS, such as continuous flushing, tachycardia and arrhythmias, bronchial wheezing, hyperthermia, peripheral cyanosis, severe diarrhea and central nervous system dysfunction. Kinney et al. first clinically termed CC as severe hypotension with the systolic blood pressure (SBP) < 80 mmHg for more than 10 min, accompanied by the presence of flushing, urticaria, ventricular dysrhythmia, bronchospasm or acidosis, and they registered 15 cases of CC among 119 patients analyzed [9]. Subsequently, Condron et al. preferred to broaden the definition to a significant hemodynamic instability (SBP < 80 or >180 mmHg, heart rate > 120 bpm) or potential end-organ dysfunction not attributable to other causes and described a much higher incidence of CC, approaching 30% in their case series [11]. Seymour et al. also included bronchoconstriction and flushing in their definition of CC [25], while Massimino et al. combined the definitions of Kinney and Condron, considering all patients with systolic blood pressure (SBP) < 80 mmHg for more than 10 min) or presentation of hemodynamic instability (including hypotension, sustained hypertension or tachycardia) [10]. As a consequence, this variability in the definition of CC leads to the impossibility of determining the real incidence and the appropriate severity of these episodes, with no conclusive studies available.

The overhang of pressure values represents a minor common denominator among all definitions. The recent study of Condron et al. tried to better characterize the underlying physiological mechanism [26]. Assuming there are three putative hormonally driven pathways of the hypotension (the reduction in cardiac output due to pulmonary artery vasoconstriction, the coronary vessels vasoconstriction resulting in cardiac failure and peripheral vasodilatation with consequent hypovolemic shock), they concluded that the pathophysiology of CC appears consistent with distributive shock. Using intraoperative, transesophageal echocardiography, pulmonary artery catheterization and intraoperative blood collection, they observed a statistically significant reduction in systemic vascular resistance among all crises. The contextual rating of hormone levels during CC exhibited markedly diverse profiles, so the Authors concluded that another carcinoid-related manifestation, such as flushing or wheezing, is not mandatory to declare a CC.

Based on the results of this study, the scientific community is now working to clarify which is the precise etiopathogenetic mechanism behind the onset of CC and which endogen molecule is the major molecule responsible for hemodynamic instability to better classify and recognize this life-threatening event and to define the best clinical approach.

## 5. Etiopathogenesis of Carcinoid Crisis

As previously cited, Kahil et al. were the first Authors who approached the topic of CC. At the end of their experience, they concluded that an excessive amount of serotonin in the bloodstream, the principal amine responsible for CS, could cause CC as a consequence of increased tryptophan intake in diet or, less likely, of a release after spontaneous necrosis of the tumor. The treatment consisted of anti-serotonin/histamine agents instead of epinephrine and levarterenol, to avoid catecholamines use that can worsen or elicit CC [2]. Tryptophan is an essential precursor of serotonin, which is hydroxylated by the rate-limiting enzyme tryptophan hydroxylase and subsequently decarboxylated by an aromatic acid decarboxylase to serotonin [27] (Figure 1). It has been demonstrated that increased tryptophan intake in diet promotes a rise in peripheral serotonin.

The vast majority of peripheral serotonin is produced by enterochromaffin cells, the same cells constituting NET masses. Therefore, the rise of tryptophan in blood circulation determines an upregulation in NET cells, with a massive production of serotonin. This hormone has a wide range of peripheral effects (Figure 2).

Concerning the cardiovascular system, a direct effect on the smooth muscle of blood vessels can lead to either vasoconstriction or vasodilatation, depending on the particular vessels influenced: renal, placental and umbilical vessels respond with vasoconstriction, whereas coronary vessels and vessels of the skeletal muscle respond with vasodilatation. The consequent effect on the blood pressure consists of three phases: first, there is a brief early depressor phase; then comes the pressor phase, as serotonin increases the total peripheral resistance; and finally, when serotonin dilates the vessels of the skeletal muscles, a late depressor phase is observed. Serotonin also influences respiration by stimulating the carotid and aortic chemoreceptors. The result is a short-lasting increase in respiratory minute volume by direct stimulation of the smooth muscles of the bronchi, leading to bronchoconstriction. Furthermore, serotonin stimulates the gastrointestinal tract to greater motility, with increased tone or intense spastic contractions, colics and evacuation of the bowels [28].

However, serotonin cannot be the unique molecule implicated in CC. Some reports denied the role of serotonin in flushing [29,30], one of the most typical symptoms associated with hemodynamic instability in CC. In addition, NETs originating from the respiratory tract and foregut do not express the aromatic acid decarboxylase, which converts 5-hydroxytryptophan to serotonin. Indeed, in this subtype of neoplasms, an atypical carcinoid syndrome can be observed characterized by patchy, sharply demarcated, serpiginous and cherry-red flushes (gastric NET) [31] or by prolonged flushing, lasting for hours or days, associated with disorientation, anxiety and tremor, lacrimation, salivation, hypotension, tachycardia, diarrhea, dyspnea, asthma and edema (Pulmonary NETs) [32]. Other potential mediators of CC are bradykinins, prostaglandins, tachykinins, substance P and histamine.

Table 2 summarizes the main humoral molecules involved in the pathogenesis of CS/CC.

In CC, the most implicated compounds seem to be vasoactive peptides, such as serotonin, histamine, bradykinin, tachykinins and kallikrein. In particular, patients with carcinoid flushing exhibit elevated levels of bradykinin and kallikrein in the bloodstream [33], which are considered the most probably responsible for flushing, rather than serotonin.

The only available data concerning the incidence and possible biological onset of CC is derived from few, small studies on intraoperative CC, such as surgical resection or locoregional treatments. When tumor manipulation occurs, the stress response triggers the release of catecholamines from the adrenals or sympathetic neurons, which in turn contribute to the release of tumor products [10]. Although CC is more typical of functional NETs, it may also occur in nonfunctional tumors. The presence of liver metastases, older age, and, for intraoperative crises, anticipated long anesthesia time, represent the most significant risk factors associated with the onset of CC, as demonstrated by Massimino et al. [10] and subsequently prospectively confirmed by Condron et al. [11]. Recently, the Research Unit of the Oregon Health and Science University has evaluated hemodynamic parameters and serum hormone levels during elective major surgery in patients with NETs [26]. Forty-six patients with a high risk of CC (older age, liver involvement, long anesthesia) were enrolled. The patients presented pulmonary artery catheters inserted to track pulmonary artery pressure, cardiac output and systemic vascular resistance, in addition to transesophageal echocardiography probes inserted to supervise cardiac function, and had serial measurements of typical hormones considered to be implicated in CC (serotonin, histamine, bradykinin and kallikrein). Seventeen patients experienced CC with prolonged hypotension. The most significant finding was that the pre-incision serotonin level was significantly higher in patients who manifested the crisis, while there were no significant changes in the mean value of any of the four hormone levels during CC. Patients manifesting CC presented an increased risk of postoperative complications, particularly if the events continued for more than 10 min. Authors concluded that elevated pre-incision serum serotonin levels represent a novel marker for increased risk of CC, as well as the severity of the crisis, and they observed no evidence of any massive release of the evaluated hormones during CC, suggesting CC may be an entirely separate pathophysiologic entity from that of CS. However, since, to date, no other study has been carried out on this topic, this hypothesis needs further confirmation.

## 6. Carcinoid Crisis Management

### 6.1. Octreotide

Assuming that CC is caused by the massive release of hormones by tumor masses, octreotide has historically represented the mainstay of its treatment. This drug is a long-acting synthetic octapeptide [34] that acts like somatostatin, a potent inhibitory peptide (Figure 3). The blockade of hormone releases, such as insulin, glucagon, gastrin and other gastrointestinal molecules, and the reduction of splanchnic and hepatic blood flow represent the mechanisms underlying the management of CC. However, the real role of octreotide in the management of CC is not well established yet, and the available data are partly contradictory.

The first report of using octreotide to treat CC dates back to 1985 [36], when Kvols et al. administered two intravenous boluses of 50 μg of this drug to a patient affected by a small bowel NET presenting life-threatening hypotension and prolonged flushing during abdominal surgery, unresponsive to intravenous fluids, intravenous calcium or intravenous conventional vasopressors. The Authors described a rapid resolution of the critical status and concluded that octreotide must be available in the operating room during surgery to rapidly manage CC. Furthermore, the same Authors conducted an explorative study on 25 patients with metastatic NET and documented CS [37] with the aim to evaluate the effect of this long-acting somatostatin analog, suggesting that this drug could be routinely safely used in the management of CS with excellent results.

However, further studies have questioned the real role of octreotide in the specific treatment of CC. Considering the pharmacodynamic profile of octreotide, this drug acts as a hormone release blocker and not as a hormone receptor blocker, therefore it should not neutralize the effect of circulating vasoactive peptides [38,39]. It should possibly prevent a worsening in the CC or prevent its occurrence at all. According to the most recent analysis put forward by Wann et al. [40], the rapid resolution of the acute episode described by Kvols et al. could also be explained by a delayed effect of epinephrine or other previously administered medications.

In 2001, Kinney et al. [9] evaluated the complication rate and outcomes of a larger series of patients with metastatic NET. Among the 119 subjects undergoing abdominal surgery included, 6 received only a preoperative octreotide dose, while 45 patients received octreotide intraoperatively, and 73 patients did not receive octreotide. A total of 15 out of 119 patients experienced perioperative complications, including 3 deaths, but none of the 45 subjects who received the intraoperative drug dose had an intraoperative complication. The researchers reported a statistically significant difference in terms of intraoperative complications between the 45 patients who received intraoperative octreotide and the 73 who did not (*p* = 0.023). They concluded that patients with metastatic carcinoid tumors can undergo abdominal surgery safely with an intraoperative octreotide dose, reporting a significantly global decrease in intraoperative complications such as CC.

Based on the possible preventive role of octreotide on CC, Massimino et al. retrospectively explored the use of octreotide prophylaxis in a group of 97 patients undergoing surgery during the years 2007–2011. A total of 87 patients (90%) received prophylactic octreotide (dose range 100–1100 mg, median 500 mg), and 56% received at least one additional intraoperative dose. Despite the use of octreotide, intraoperative complications occurred in 23 (24%) patients. Therefore, the obtained data greatly diverged from the results published by Kinney et al., In their series, 18 patients (19%) experienced prolonged hypotension, while 5 (5%) were reported to have had marked hemodynamic instability consistent with a CC. Intraoperative complications occurred with the same frequency among patients with functioning (21%) and non-functioning (28%) NET, and the presence of liver metastases was found to be a predictor of intraoperative complications. These findings suggest neither outpatient octreotide LAR nor single-dose preoperative bolus octreotide prevent all intraoperative complications [10].

In 2016, Woltering et al. [41] suggested a possible solution to improve octreotide effectiveness. Their retrospective report demonstrated a reduction in CC incidence by using a continuous infusion of high-dose octreotide during surgery. As anesthetic or surgical stimuli can potentially precipitate an unpredictable release of amines, in their protocols, the researchers administered a prophylactic preoperative 500 μg bolus of octreotide acetate along with a continuous intravenous intraoperative infusion for all NET patients undergoing surgical cytoreduction, regardless of the location of their primary tumor or their functional status. The rationale behind this choice is connected to octreotide pharmacokinetics: preoperative bolus of octreotide, with a half-life of 90–120 min, might not last long enough for protection against CC during long surgery. Without a continuous infusion, blood level would fall to 50% of the original octreotide concentration after 2 h and would be only 25% of the original concentration after 4 h. As result, Woltering et al. reported an incidence of CC of only 3.6%, concluding that continuous intraoperative octreotide infusion could significantly reduce the risk of CC onset.

To demonstrate the benefits of continuous octreotide infusion on CC prevention, Condron et al. [11] prospectively enrolled 127 patients (71% with liver metastases, 74% with CS) who underwent 150 operations with continuous octreotide infusions. Contrary to what was expected, 30% of the patients manifested CC, and the crises were significantly associated with the presence of liver metastases (*p* = 0.02) or a history of CS (*p* = 0.006). The rapid use of vasopressor was effective in reducing crisis duration, with a reduction in postoperative complications. These unexpected results could be explained by analyzing CC definitions. While Woltering et al. considered only episodes of hypotension lasting ≥10 min [41], Condron et al. registered all cases of hemodynamic instability (systolic blood pressure < 80 or >180 mmHg, heart rate > 120 beats per minute or display of any physiology that, if sustained, would be expected to lead to end-organ dysfunction, such as ventricular arrhythmias or bronchospasm) unattributable to any other causes [11]. Considering only episodes of hypotension lasting ≥10 min, as Woltering et al. did, Condron et al. would have reported only 8% of CC.

Other reports on the outcome of prophylactic octreotide were published in 2018 [42] and 2019 [12]. Kinney et al. retrospectively evaluated 169 patients undergoing partial hepatic resection for metastatic NET between 1997 and 2015, and 77% (130/169) of patients preoperatively received 500 μg of subcutaneous octreotide. In their report, there were no documented cases of CC; one patient developed clinical findings of an emerging CC but was successfully treated with doses of octreotide, and findings resolved in <10 min. Of note, in this case, CC was defined as a sudden or blunt onset of at least two of the following: flushing or urticaria that are not explained by an allergic reaction; bronchospasm or bronchodilator administration; hypotension (SBP < 80 mmHg for >10 min and treated with vasopressor) not explained by volume status or hemorrhage; tachycardia ≥120 beats per minute [42]. Analyzing only sustained hypotension, the incidence was 5.6%, but because none of those patients exhibited any other criteria, none were considered CC.

Finally, Kwon et al. reported a retrospective series of 75 patients with metastatic well-differentiated NETs who underwent liver resection, ablation or embolotherapy from 2012 to 2016. The CC was defined subjectively by clinical documentation of occurrence by any treating physician, including the anesthesiologist, surgeon or interventional radiologist, and it had to be associated with hemodynamic instability, defined as the presence of at least one of the following events sustained for more than 10 min during the procedure: hypotension (systolic blood pressure, 80 mm Hg) or tachycardia (heart rate, >120 beats per minute). CC was identified in 32% (24) of patients. Of note, the route and dose of preprocedural octreotide administration varied widely. Neither long-acting octreotide, perioperative octreotide, intraoperative octreotide nor any combination was associated with a lower incidence of crisis. The Authors concluded that somatostatin analogs do not reliably prevent CC. One hypothesis is that CC may be a phenomenon physiologically distinct from CS, involving the release of a different distribution of vasoactive substances, against which different therapeutic agents can be used [12]. As previously mentioned, this proposal has been explored by Condron et al. [26].

Data are summarized in Table 3.

As reported in the most recent guidelines referring to clinical studies discussed so far, there are no standard octreotide regimens in the management of CC; subcutaneous administration of octreotide 100–200 mcg × 2–3 daily during surgery has been suggested for a minor procedure or lower-risk patients. However, intravenous octreotide infusions should also be readily available in the operating room to be used when deemed necessary. For major surgery, perioperative prophylactic treatment with intravenous octreotide, at the starting dose of 50–100 mcg/h (mean dose 100–200 mcg/h), is the standard regimen used by most clinical centers. Although this has not been substantiated by any prospective study, most experts start treatment with intravenous octreotide 12 h before the operation and escalate the dose as necessary until symptom control is achieved. This infusion continues for at least 48 h after the operation, with dose titration as clinically required [43].

Considering the increasing use of PRRT, several studies have been conducted to estimate the impact of octreotide in PRRT-induced CC. The incidence of CC during PRRT ranges between 1 and 10%, and, as previously mentioned in this review, specific risk factors have been defined, which, if present, expose the patient to a major rate of intra-treatment complications [24]. In the clinical experience reported by de Kaizer et al., among 479 patients enrolled in the study, 7 cases of CC after the first cycle of PRRT were reported [44]. The treatment included high-dose octreotide, i.v. fluid replacement and corticosteroid. Despite additional precautions taken after their first therapy cycle (continuation of somatostatin analog, corticosteroids and reduction of administered dosage of ^177^Lu-octreotate), 3 patients developed a second CC after the subsequent cycle of PRRT. Analyzing the possible mechanism behind hormonal secretion during Lu-octreotate therapy (tumor lysis vs. discontinuation of short-acting somatostatin analog vs. emotional stress response to hospitalization vs. administration of arginine and lysine), the Authors concluded that hormonal crises should be managed with an infusion of somatostatin analogs i.v., fluids, corticosteroids and correction of electrolyte disturbances. More recently, Stenzel et al. came to the same conclusions [45]. Moreover, the Australian group of Tapia Rico et al. first proposed a protocol to prevent and manage severe CC and specifically defined which patients would benefit most (“high-risk patients”) from premedication to PRRT therapy with corticosteroids and bolus dose of octreotide sc [46]. This work has been further enriched by De Olmo et al., who recently published the current procedure adopted for approaching patients undergoing PRRT [24]. The work of clinicians should start from the identification of risk factors for CC through:-The evaluation of nutritional assessment with the diagnosis and correction of hydro electrolytic disorders, malnutrition and malabsorption, and the avoidance of food triggers and intensive physical exercises the previous day;-The evaluation of NET characteristics (high tumor burden, use of somatostatin analogs to control CS).

In particular, in the case of bulky tumors, it is mandatory to consider a possible surgery or locoregional treatment before PRRT. In addition, it is essential to obtain good control of CS before the first cycle of ^177^Lu-octreotate.

As premedication, the Authors advise the administration of corticosteroids (dexamentasone 4–8 mg), antiemetic (ondansetron 4 mg), somatostatin analog (octreotide 100 μg s.c. or 50 μg i.v. bolus), antistaminic h1 (dexchlorpheniramine 5 mg i.v. in slow infusion) and antistaminic H2 (ranitidine 50 mg i.v. in slow infusion). In the event of an outbreak of CC, the infusion of Lu-DOTATATE should immediately stop, and a bolus of octreotide (100–500 μg) should be immediately administered, followed by a maintaining dose of 50–100 μg/h infusion. In case of severe hypotension, the Authors do not exclude the use of phenylephrine or vasopressor drugs.

### 6.2. Vasopressors

Based on clinical experiences [2,13,14,47], sympathomimetic drugs have always been avoided in the management of CC-related hypotension, since they may worsen it by triggering further release of peptides by tumor masses. Mason et al., in 1966, first observed vascular response to systemic epinephrine injection in the forearm in 7 NET patients. The Authors noted decreased systemic blood pressure, decreased vascular resistance and an elevated bradykinin level for 5 min after injection, although the response was quite variable among the 7 patients [47]. However, data supporting a widespread concern about the abuse of beta-adrenergic agonists remain very limited [48].

More recent experiences have tried to explore the real role of this class of drugs in the management of intraoperative CC [10,26,48]. Notably, Limbach et al. retrospectively examined the use of vasoactive medications during CC to determine whether an association between induction of a secondary CC and beta-adrenergic agonist administration could be detected. They observed no close correlation between the use of sympathomimetic drugs and secondary CC and, in addition, the duration of CC did not increase with administration of beta-adrenergic agonists [48].

Based on these reports and considering the recent findings, CC-related hypotension is due to a distributive shock, as previously discussed, and the administration of vasopressors, which determine a wider systemic effect (vs. octreotide alone, which has a vasoconstrictive effect only on splanchnic vessels), should be suggested when standard measures fail [40].

## 7. Conclusions

The CC remains a severe, sometimes fatal, acute manifestation of NETs. The rarity and unpredictability of this event has hampered the development of randomized controlled trials to define the best clinical and therapeutical approach. Data available in the literature are mainly derived from small retrospective studies or case reports, in which the clinical definition of CC is not consistent and universally accepted. Recently, the pathophysiological mechanism has also been questioned. According to some Authors, CC should be considered a completely separate event from CS, as it can be triggered by a different cocktail of hormones.

Current guidelines, such as the European ENETS and the American NANETS, continue to recommend the administration of octreotide to prevent and manage CC onset [42,49], although without a standard scheme, duration and dose specification. Furthermore, the use of vasopressors has been recently revised, as their administration during crises could accelerate the increase in blood pressure.

Based on recent evidence, some institutes stopped using octreotide during operations altogether instead of relying on vasopressors, including beta-adrenergic agonists [50]. Their experience on 195 patients demonstrates a rate of CC not significantly higher than that reported in previous studies. The Authors conclude that perioperative octreotide use may be safely stopped, owing to inefficacy, and the treatment of crisis should be replaced with intravenous fluids and vasopressors, which address the actual pathophysiology of the crisis, without concern about increasing crisis duration or rates of major postoperative complications.

Further efforts should be directed toward the understanding of the correct pathophysiology of CC to propose more specific, effective and established treatments.

## Figures and Tables

**Figure 1 cancers-14-00662-f001:**
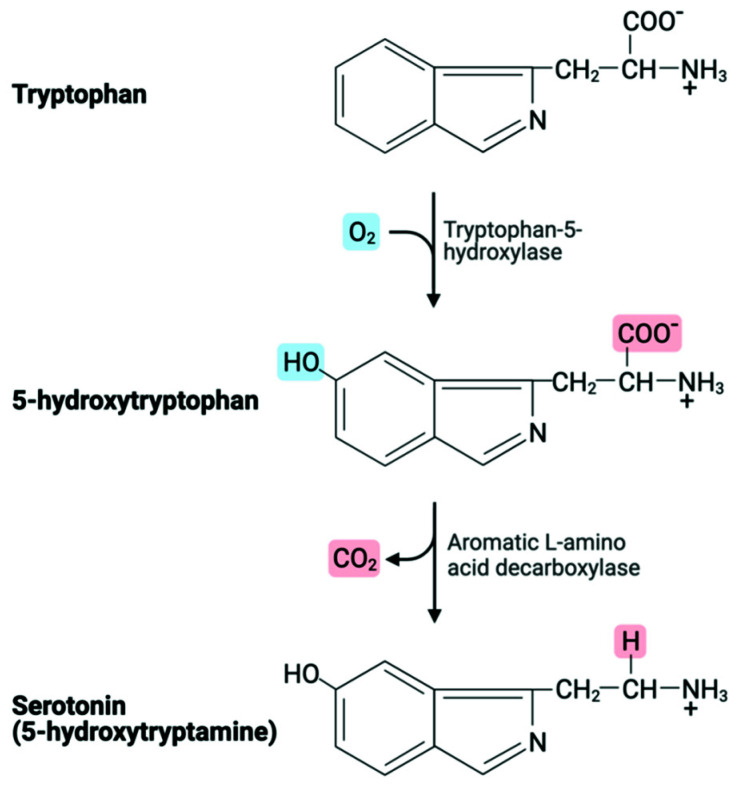
Tryptophan metabolic pathway.

**Figure 2 cancers-14-00662-f002:**
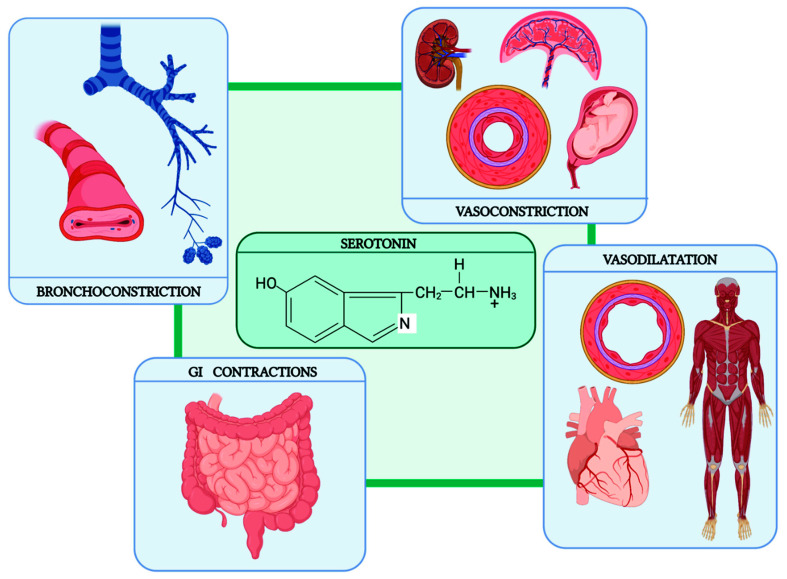
Peripheral effects of serotonin.

**Figure 3 cancers-14-00662-f003:**
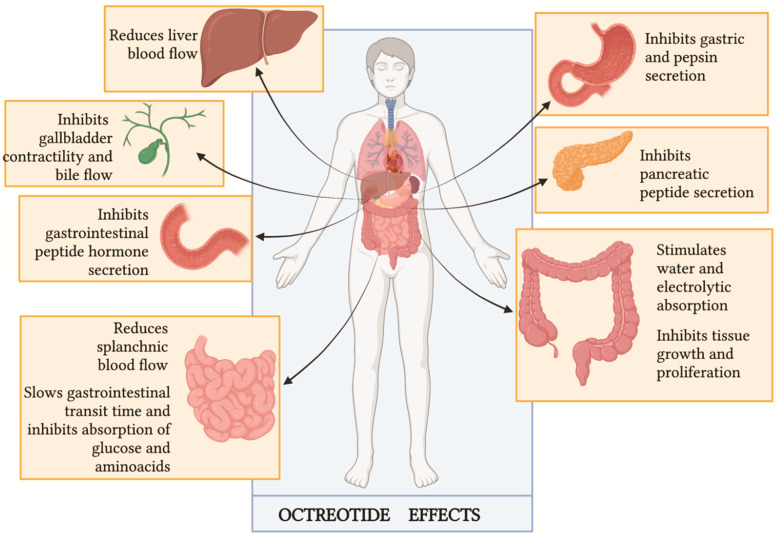
Octreotide clinical effects (Figure adapted by Lamberts SW, van der Lely AJ, de Herder WW, Hofland LJ. Octreotide [35]).

**Table 1 cancers-14-00662-t001:** Case reports of Carcinoid Crisis up to September 2021.

Authors and Date	Primary Tumor Location	Clinical Presentation	Triggering Factor	Treatment
Kahil et al., 1964 [2]	ileum	apprehension, chest pain, abdominal cramps, diarrhea, flushing, cyanotic extremities, hypotension	increased tryptophan intake in diet	metaraminol, levarterenol (ineffective), cyproheptadine
Harris AL et al., 1983 [13]	ileum	prolonged continuous flushing, confusion, hypotension, coma	ileotransverse colostomy and liver biopsy	anti-serotonin and antikinin agents (5 fluorouracil, trasylol, prednisone, cimetidine, cyproheptadine, methysergide, tryptophan, aminoplex 12)
Hughes et al., 1989 [14]	lung	hypertension, tachycardia	anesthesia induction	ketanserin, octreotide
Batchelor AM et al., 1992 [15]	lung	peripheral cyanosis, myocardial infarction, flushing	rigid bronchoscopy	adrenaline, hydrocortisone, octreotide, ketanserin
Parry R.G. et al., 1996 [16]	hepatic metastases	acute tubular necrosis oliguria, diarrhea, flushing	liver biopsy	glucocorticoids, hemodialysis, octreotide, cyproheptadine
Koopmans KP et al., 2005 [17]	ileum	hypertension, peripheral cyanosis, flushing, edema, vomiting	^18^F-DOPA infusion during PET	antihistamine
Papadogias et al., 2007 [18]	lung	hypotension, diarrhea	radioembolization (^111^in-octreotide infusion via intra-arterial injection)	octreotideic, alpha-interferon, glucocorticoids, and H1–H2 histamine receptor blockers
Van Diepen et al., 2013 [19]	small bowel	hypotension, fever, flushing	valve replacement	octreotide, vasopressin, norepinephrine, hydrocortisone, anti-serotonin, antihistamine, cyproheptadine
Kromas ML et al., 2017 [20]	lung	hypotension, wheezing	bronchoscopy	octreotide bolus
Maddali MV et al., 2020 [21]	ileum	initial hypertension and tachycardia, followed by shock and respiratory failure	TACE	dobutamine and vasopressin, then milrinone and nitroprusside (ineffective), octreotideic
Dhanani et al., 2020 [22]	small bowel	hypotension, loss of consciousness, cardiac arrest	Peptide Receptor Radionuclide Therapy (PRRT)	cardiopulmonary resuscitation plus adrenaline (ineffective), octreotideic
Mahdi et al., 2021 [23]	transverse colon (NEC)	abdominal pain, hypotension	not mentioned	empiric antibiotic therapy, norepinephrine ic (ineffective), octreotideic

**Table 2 cancers-14-00662-t002:** Molecules implicated in CS/CC.

	Effects	Role in CC/CS
Amines		
Serotonin	vasoconstriction/vasodilatation,	diarrhea, cramps
	bronchoconstriction,	bronchospasm
	fibroblastic activation	carcinoid heart disease
Histamine	vasoconstriction/vasodilatation	flushing, pruritus, edema
	bronchoconstriction	bronchospasm
	tachycardia	
5-Hydroxytryptophan	vasodilatation	diarrhea, cramps
Norepinephrine	vasoconstriction, tachycardia, hyperglycemia, hyperlipidemia, tremor	anxiety
Dopamine	vasodilatation, GI motility block	
**Polypeptides**		
Kallikrein	conversion of kininogens in kinins (bradykinin and kallidin)	flushing, bronchospasm
Bradykinin	vasodilatation, bronchoconstriction, edema	flushing, bronchospasm
Somatostatin	GH, TSH, prolactin, insulin, glucagon release inhibition	diabetes, cholelithiasis, steatorrhea, hypochloridria
Motilin	GI motility stimulation	diarrhea, cramps
Pancreatic Polypeptide	pancreatic secretion regulation (inhibits the secretion of fluids, bicarbonate, and digestive enzymes)	
Vasoactive Intestinal Peptide	vasodilatation, smooth muscle relaxation induction, secretion of water into pancreatic juice, and bile stimulation	profuse diarrhea, hypokalemia, achlorhydria
Neuropeptide K (tachykinin family)	bronchoconstriction, bradycardia	
Substance P (tachykinin family)	bronchoconstriction	
	bradycardia	
Neurokinin A (tachykinin family)	bronchoconstriction	
	bradycardia	
Neurokinin B (tachykinin family)	bronchoconstriction	
	bradycardia	
Corticotropin (ACTH)	cortisol release	Cushing Syndrome
Gastrin	hydrochloric acid release by the stomach	Zollinger Ellison Syndrome
Growth Hormone	cell metabolism stimulation	acromegaly
Peptide YY	anorectic effect	
Glucagon	glucose and fatty acid release	necrolytic migratory erythema, weight loss hyperglycemia
Beta-endorphin	pain relief	
Neurotensin	gastrin and motilin release inhibition, vasodilatation	
Chromogranin A	vasostatin precursor, pancreastatin, catestatin, and parastatin that inhibit hormone released by neuroendocrine cells	
**Prostaglandins**	vasoconstriction/vasodilatation	

**Table 3 cancers-14-00662-t003:** Articles assessing the impact of octreotide in CC.

Variation	Type of Paper	Number of Patients	Number of CC	Octreotide Dose and Regimen
Kvols et al., 1986 [36]	Case report-retrospective study	25	1	a bolus of 50 μg of octreotide intraoperatively
Kinney et al. [9]	Retrospective study	119	15 (none of the pts received onctreotide intraoperatively)	- 31 pts received octreotide preoperatively (median dose 300 μg—range 50–1000 μg); 25 of these pts received additional octreotide intraoperatively. - 45 pts received octreotide intraoperatively (median dose 350 μg—range 30–4000 μg)
Massimino et al. [10]	Retrospective study	97	23	87 pts received prophylactic octreotide (median dose 500 μg—range 100–1100 μg) + intraoperative bolus if necessary (median dose 350 μg—range 100–5500 μg)
Woltering et al. [41]	Retrospective study	150	6	Continuous high-dose octreotide infusion: 500 μg/h
Condron et al. [11]	Prospective study	127	38	Continuous high-dose octreotide infusion: 100 μg/h
Kinney et al. [42]	Retrospective study	169	0	- 130 pts received 500 μg preoperatively s.c. - 39 pts received additional intravenous octreotide (median dose 500 μg—range 250–650 μg)
Kwon et al. [12]	Retrospective study	75	24	- 27 pts received preprocedure infusion (median dose 150 μg/h—range 50–300 μg/h) - 21 pts received a preprocedure i.v. or s.c. bolus (median dose 150 μg—range 100–300 μg). - 48 pts received intraprocedural infusion (median dose 150 μg/h—range 50–300 μg/h) - 20 pts receive an intraprocedural i.v or s.c. bolus (median dose 150 μg—range 20–510 μg)

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
