# Peer review of "Carcinoid Crisis: A Misunderstood and Unrecognized Oncological Emergency"

_cancers, 2022, doi:10.3390/cancers14030662_

Round 1

Reviewer 1 Report

The authors describe the ethiopathogenesis, definition and clinical manifestations and management of carcinoid crisis. Although interesting, I think the review should be reorganized in a more understanding and practical way. Major revision is needed.

Other commentaries:

-avoid using "we" when writing, better the impersonal way-

-English has to be revised by a native speaker.

- if image are not original indicate

-although carcinoid crisis can appear in several circumstances it seems that for the authors all the evidence is on perioperative management, specific comments should be made on other procedures such as liver embolization or treatment with radioligand therapy; and include all these in the evidence; if not, the review has to specify that the review is only in surgical procedures.

- it is also mentioned on lines 47-48 that it usually appears on foregut and midgut...the most common is bronchial and intestinal.;.pancreas, gastric, duodenum, colon are very rare and this should be highlighted in the text. Revise and correct.

-ETHIOPATHOGENESIS: the first case described in the literature should be commented but it is too long--> shorten this part.

-CLINICAL MANIFESTATIONS AND DEFINITION:  the review should start with the definition and clinical manifestations and proceed with the ethiopathogenesis. First define carcinoid syndrome and carcinoid crisis with the literature evidence and clinical manifestations. It has no sense to talk about something you haven’t defined before.

Curiously table 2 has a lot of studies on carcinoid crisis and treatment approach is in the table but no reference to this is made on the text in the part of management. Revise and correct.

-MANAGEMENT: The most interesting part is the management of carcinoid crisis in which there is more controversy the previous parts should be shortened and this part should be the main document. A systematic review in this part is recommended. A table with the management and prevention of the studies commented and with their characteristics should be made to clarify…

Author Response

Revisor 1. We reorganized the review in a more understanding way:
1. We switched to impersonal structure of sentences
2. We revised English expressions
3. We indicated which Image was adapted by another paper
4. We add a whole paragraph discussing about other causes of carcinoid crisis
5. We highlighted associations between primary tumor location and incidence of carcinoid crisis
6. We shortened the paragraph about the first case report described in literature
7. We inverted the order of paragraphs "clinical manifestation" and "ethiopathogenetic", explaining clearly the definition of Carcinoid Crisis
8. In the section "Management" We added a resuming table with all the studies mentioned in the text and we expanded treatment and prophylaxis paragraph.

Reviewer 2 Report

The paper entitled “Carcinoid Crisis: a Misunderstood and Unrecognized Oncological Emergency” discussed the Carcinoid Crisis related to neuroendocrine tumors. Specifically, the potential etiopathogenetic mechanisms, clinical implications, and potential treatments were reviewed. This is an excellent review paper. There are some minor scientific questions that need be addressed. The detailed scientific review comments are listed below.  

  • There are several approved other anti-NET therapies, such as PRRT (Lutathera) that combines endoradiotherapy (177Lu-DOTA-TATE). Is there any report of CC management post PRRT treatment?
  • Is there any other strategies to manage CC in clinics?

Author Response

Revisor 2:  Thank you for your appreciations. As you suggested, we talked about PRRT, underlining possibile therapeutic strategies.

Round 2

Reviewer 1 Report

I think the article has improved considerably, there are several typographical errors to check